# Integrase Strand Transfer Inhibitors Are Effective Anti-HIV Drugs

**DOI:** 10.3390/v13020205

**Published:** 2021-01-29

**Authors:** Steven J. Smith, Xue Zhi Zhao, Dario Oliveira Passos, Dmitry Lyumkis, Terrence R. Burke, Stephen H. Hughes

**Affiliations:** 1HIV Dynamics and Replication Program, Center for Cancer Research, National Cancer Institute, Frederick, MD 21702, USA; steven.smith3@nih.gov; 2Chemical Biology Laboratory, Center for Cancer Research, National Cancer Institute, Frederick, MD 21702, USA; xuezhi.zhao@nih.gov (X.Z.Z.); burkete@mail.nih.gov (T.R.B.J.); 3Laboratory of Genetics, The Salk Institute for Biological Studies, La Jolla, CA 92037, USA; dariorj@salk.edu (D.O.P.); dlyumkis@salk.edu (D.L.); 4Department of Integrative Structural and Computational Biology, The Scripps Research Institute, La Jolla, CA 92037, USA

**Keywords:** HIV, integration, INSTIs, drug resistance, antiviral therapy

## Abstract

Integrase strand transfer inhibitors (INSTIs) are currently recommended for the first line treatment of human immunodeficiency virus type one (HIV-1) infection. The first-generation INSTIs are effective but can select for resistant viruses. Recent advances have led to several potent second-generation INSTIs that are effective against both wild-type (WT) HIV-1 integrase and many of the first-generation INSTI-resistant mutants. The emergence of resistance to these new second-generation INSTIs has been minimal, which has resulted in alternative treatment strategies for HIV-1 patients. Moreover, because of their high antiviral potencies and, in some cases, their bioavailability profiles, INSTIs will probably have prominent roles in pre-exposure prophylaxis (PrEP). Herein, we review the current state of the clinically relevant INSTIs and discuss the future outlook for this class of antiretrovirals.

## 1. Introduction

Integrase strand transfer inhibitors (INSTIs) are important drugs that are currently used in efforts to control human immunodeficiency virus type one (HIV-1) infections and limit the spread of the virus (Figure 1). We will briefly consider the discovery, efficacy, mechanism of action, and resistance to the clinically relevant INSTIs. Before the inception of effective antiretroviral therapy (ART), HIV-1 infection was a death sentence. The development of ART changed the prognosis to a chronic illness that can, in most cases, be managed and controlled [1]. This progress is due to the discovery and development of effective and well-tolerated anti-HIV-1 drugs. Currently, ART usually involves the combinations of three anti-HIV-1 drugs (Appendix A) [2]. The antiviral drugs most commonly recommended for ART include an INSTI (Figure 1) and two nucleoside/nucleotide reverse transcription (RT) inhibitors (NRTIs), or, in some cases, an INSTI, an NRTI, and a nonnucleoside RT inhibitor (NNRTIs). In most compliant individuals, ART completely blocks viral replication, which prevents the emergence of drug resistance [3]. The latest FDA-approved INSTIs have proven to be so effective that attempts are being made to reduce the treatment from the standard three-drug cocktail to a two-drug regimen (Appendix A) [4,5]. Moreover, the potency and tolerability of the available INSTIs has led to dosing strategies in which INSTIs are formulated into intravaginal rings, injectables, or implants from which the drugs are slowly released at controlled rates so that the concentrations of the drugs that inhibit HIV-1 infection are maintained for weeks or months [6,7,8,9,10]. In addition, some INSTIs have the potential to be administered as pre-exposure prophylaxis (PrEP) agents to individuals who are most likely to contract HIV-1 [11]. The data from recent PrEP trials are promising [12].

There are still problems with HIV-1 developing resistance to INSTIs and to other anti-HIV-1 drugs. It is relatively easy for HIV-1 to develop resistance to the first-generation INSTIs, raltegravir (RAL) and elvitegravir (EVG) [13,14]. However, the second-generation INSTIs dolutegravir (DTG) and bictegravir (BIC) are able to effectively inhibit most of HIV-1 IN mutants that arise in response to RAL and EVG (Figure 1) [15,16,17]. Although there have been no reports of the development of resistance to DTG and BIC in treatment-naïve individuals who have been put on ART regimens that include these INSTIs [18,19,20,21,22], there has been a poor response to salvage therapies that included DTG in some individuals who had previously failed ART regimens that included a first-generation INSTI [16,17]. In addition, in vitro experiments have identified combinations of mutations in IN that markedly reduce the potencies of both DTG and BIC [15,23,24,25].

The successful use of INSTIs in the clinic has been accompanied by progress in understanding the mechanism of action of INSTIs at both the biochemical and structural level. There are structural data which show how INSTIs bind in the active sites of HIV-1 and simian immunodeficiency virus (SIV) red-capped mangabey (rcm) INs [26,27]. These new data confirmed the mechanism of inhibition by INSTIs that was discovered using structures of the prototype foamy virus (PFV) intasome [28,29]. Importantly, because there are differences in the structures of PFV IN and HIV-1 IN, the new structural information will help in the ongoing efforts to improve the efficacy of INSTIs against drug-resistant HIV-1 IN mutants.

## 2. Background

HIV-1 encodes three enzymes, protease (PR), reverse transcriptase (RT), and integrase (IN). Each of the three enzymes perform essentials steps in the viral lifecycle. PR is involved in cleaving the Gag and Gag-Pol polyproteins to produce the mature forms of the structural proteins and the three viral enzymes, PR, RT, and IN [30]. RT, which has two enzymatic activities (a polymerase and an RNase H), converts the single-stranded viral RNA found in the virion into double-stranded linear DNA in a newly infected host cell. IN integrates the linear viral DNA into the host genome [31]. IN carries out two reactions, both of which involve the same active site. In the first step, 3′-processing (3′-P), a GT dinucleotide is removed from both of the 3′-ends of the linear viral DNA that was produced by RT [32,33]. The 3′-P reaction leaves a conserved CA dinucleotide at each of the processed 3′-ends of the viral DNA. In the second step, strand transfer (ST), IN catalyzes a reaction in which the hydroxyl groups of the dAs at the 3′ ends of the processed viral DNA act as nucleophiles to attack the host genome, causing an exchange reaction, in which the 3′-ends of the viral DNA are inserted into the host DNA [34,35]. The geometry of the exchange reaction creates a short duplication of the host sequences that flank the integrated viral DNA [36,37]. The integration reaction leaves a nick and a short mismatch with the host DNA, which are repaired by host DNA repair enzymes [38,39]. This process leads to the formation of the stably integrated form of the viral DNA, which is called a provirus.

Each of the viral enzymes has been targeted by drugs that have been used to treat HIV-1 infections. RT inhibitors were the first drugs to be approved for use in people infected with HIV-1 and have played important roles in ART for decades [40,41]. PR inhibitors [42,43,44], which played a key role in the first successful ART combinations [45], have a less important role now that INSTIs are available. IN inhibitors have been in development since the 1990s [46,47,48,49,50,51], culminating in the approval of RAL by the FDA in 2007 [52,53]. Three additional INSTIs have been approved by the FDA: EVG in 2012 [54], DTG in 2013 [55,56], and BIC in 2018 [25,57]. A fifth INSTI, Cabotegravir (CAB, Figure 1) [7,8,58,59], is currently licensed for use in Canada [60]. The available IN inhibitors are called integrase strand transfer inhibitors (INSTIs) because they selectively block the second reaction step (ST reaction) catalyzed by IN [37,50,61]. In the early 1990s, in vitro IN assays were developed in which recombinant IN protein was used to insert DNAs that mimicked the ends of viral DNA into a DNA target [62,63,64]. These in vitro integration assays were used to screen large libraries of small molecules to identify IN inhibitors. By the early 2000s, both Merck and Shionogi independently discovered that “di-keto” containing compounds could inhibit integration, and that these compounds were selective for the ST reaction [65,66,67,68]. Testing in viral replication assays showed that the 4-aryl-2,4-diketobutanoic acids (DKAs) were effective antiviral agents. Using these initial compounds as leads, additional rounds of optimization led to the discovery of compounds that potently inhibited HIV replication in cell-based assays. Those early findings paved the way for the first clinically relevant INSTIs.

## 3. First-Generation INSTIs

### 3.1. Raltegravir (RAL)

The development of RAL, the first INSTI to be approved by the FDA to treat HIV-1 infections (2007) [53], was the culmination of optimizations that can be traced back to two different classes of compounds that were studied independently by groups at Merck working to develop inhibitors of HIV-1 IN and the hepatitis C virus (HCV) polymerase [66,69]. A class of compounds, dihydroxypyrimidine carboxamides, which was initially identified by the HCV polymerase group, was shown to be a potent inhibitor of HIV-1 replication that selectively inhibited the IN ST reaction [70]. Initial optimizations of this class of compounds showed the importance of a fluorobenzyl moiety and the 2′-dimethyl substituent that were appended to the pyrimidine core [71,72]. In separate experiments, the HIV-1 IN group showed that compounds based on a *N*-methylpyrimidone scaffold potently inhibited IN ST activity [73]. Attempts to combine the best features of the two classes of compounds did not produce an improved IN inhibitor. However, the *N*-methylpyrimidinone scaffold was used to make additional derivatives; the inclusion of an oxalamide substituent was particularly helpful. Ultimately, the amine portion of the scaffold was modified to substitute an oxadiazole group for the oxalamide, producing RAL [53]. As stated above, RAL potently inhibited IN strand transfer in both in vitro assays (IC_50_) and had antiviral activity against HIV-1 in single round replication assays (EC_50_) [53,74,75]. RAL was also broadly active against a number of HIV-1 isolates, human immunodeficiency virus type two (HIV-2), and SIV [76]. In the initial studies, RAL was found to be non-inferior to efavirenz (EFV) in clinical trials that involved treatment-naïve patients (as is discussed below, subsequent studies showed RAL to be superior to efavirenz) [77]. RAL was also shown to be useful in clinical trials with treatment-experienced subjects who were infected with triple-class-resistant HIV-1 [78]. Unfortunately, like the drugs in the other classes of antiretrovirals that are used to treat HIV-1 infections, RAL is susceptible to the emergence of viral resistance [52,79]. Resistance to RAL occurs primarily through three amino acid substitution pathways, Y143, Q148, and N155 [80]. There are, in each of the three pathways, distinctive secondary amino acid substitutions that accompany the primary substitutions [80]. In all cases, the primary mutations that confer resistance also reduce the ability of the virus to replicate [15,81,82]. In some cases, the secondary mutations may not help the HIV-1 variant to evade RAL, but rather enhance the ability of the mutant virus to replicate. A list of RAL-resistant mutants seen in clinical trials, selected in cell culture, or identified in single-round replication assays, is given in Table 1.

The effects of mutations on the susceptibility of the IN mutants to RAL can be measured in single round infectivity assays. When the amino acid substitutions Y143R, N155H, and G140S/Q148H were introduced in HIV-1 pNL4-3 and the EC_50_ values of the WT and mutant viruses were determined using single round replication assays, the IN mutants Y143R and N155H caused substantial losses in potency (for Y143R, the fold change (FC), which is the decrease in potency between WT and IN mutant, was 41, and for N155H, the FC was 38). RAL displayed an even larger loss in potency with the IN double mutant G140S/Q148H (the FC was 475) [75]. Mutations in the three amino acid substitution pathways arise at different rates [81,83,84,85]. The N155H mutation often emerges first; viruses that use the pathways defined by substitutions at Y143 or Q148 usually emerge more slowly. Under continued drug pressure in vivo, additional mutations can be selected. These three amino acid substitution pathways are mutually exclusive, at least when the mutations are selected in vitro in cell culture or in vivo in patients.

Single round infection assays measuring EC_50_ values of RAL against IN double and triple mutants that comprise combinations of the amino acid substitutions from these three pathways have displayed large losses in potency against RAL [81,83,85]. However, the single round infectivity of the cross-pathway IN double and triple mutants is low: Y143H/N155H (13.7 ± 3.3), Y143R/Q148H (2.6 ± 0.2), Y143R/N155H (33.9 ± 6.5), Q148H/N155H (7.3 ± 2.9), and Q148R/N155H (13.8 ± 4.1). This could explain why resistant viruses evolve mutations within a defined pathway [15]; it is likely that the effect on viral fitness is the reason that these pathways appear to be mutually exclusive (fitness/replication capacity is discussed in a later section).

The crystal structure of RAL bound to the active site of the PFV IN in a complex with viral DNA (intasome) was solved in 2010. This structure explained how INSTIs inhibited the ST reaction (Figure 2) [28,70]. The “di-keto” motif chelates the two Mg^2+^ cofactors bound in the IN active site. The halobenzyl group hydrophobically π-π stacks with the base of the penultimate cytosine near the 3′ end of the viral DNA strand that would have integrated into host DNA, displacing the terminal adenine at the 3′ end of the viral DNA. The oxazole ring that is appended to the pyrimidine core of RAL hydrophobically π-π stacks with Y143. Thus, RAL binding blocks the active site of HIV-1 IN, displaces the 3′ end of the viral DNA, and prevents the binding of the host DNA, which effectively inhibits the ST reaction. Although RAL is vulnerable to the development of resistance, it is a better therapeutic option than many drugs from other antiretroviral drug classes. For example, RAL displayed higher efficacies and superiority when compared to EFV in the STARTMRK clinical trial (VL < 50 copies/mL in 71% of RAL patients to 61% in EFV patients) when tested in randomized treatment-naïve patients [77]. Because RAL has a favorable pharmacological profile, is well tolerated, has minimal potential for protein–protein interactions, few adverse effects, and a high efficacy against HIV-1, it is still recommended in certain clinical settings, for example, in treatment-naïve pregnant women in their third trimester (Appendix A) [86,87]. RAL can also be administered to adolescent and pediatric patients [88].

### 3.2. Elvitegravir (EVG)

A collaborative effort between Japanese Tobacco LLC. (Tokyo, Japan) and Gilead Sciences (Foster City, CA) led to the development of EVG [89]. After searching for and designing new bioisosteres of the diketo acid core, a 4-quinolone-3-carboxylic acid, which was previously used in antibiotics, was shown to inhibit the ST reaction. Modifications to this scaffold, including the addition of a halogenated benzyl moiety and the addition of a hydroxyethyl group to the 1-position on the quinolone ring, resulted in improved antiviral activity. The addition of an isopropyl group at the 1S-position of the hydroxyethyl modification improved the potency against HIV-1 (subnanomolar EC_50_ value) and led to the synthesis of EVG [54]. EVG was initially approved as part of a fixed-dose combination therapy called Stribild, which includes emtricitabine (FTC), tenofovir disoproxil fumarate (TDF), and cobicistat, the latter acting as a pharmacokinetic booster to reduce the metabolic breakdown of EVG (Appendix A) [90]. In 2014, the FDA approved EVG for use in combination with a ritonavir-boosted HIV-1 protease inhibitor and another antiretroviral drug, such as didanosine [91]. Although EVG is a first-generation INSTI, and is susceptible to the development of resistance, it is still recommended in certain clinical settings (Appendix A) [2].

EVG inhibits the HIV-1 IN ST reaction in vitro and displays potent antiviral activities against laboratory strains and clinical isolates of HIV-1 in single round replication assays [74,75,92,93]. Importantly, EVG inhibits the RAL-resistant IN mutant Y143R because it lacks the chemical moiety, present in RAL, that stacks with the phenol group of Y143. However, EVG does not potently inhibit the other well-known RAL-resistant IN mutants N155H and G140S/Q148H [74,75]. Moreover, there are additional mutations that can cause resistance to EVG. IN mutants selected by EVG in vitro include H51Y, T66I, E92Q, F121Y, S147G, Q148R, S153Y, E157Q, and R263K [93,94,95]. Clinical trials showed that some of the mutants that were selected in vitro were associated with resistance in patients: T66I, E92Q, Q148R, and N155H. Mutations that were selected less frequently in vivo included H51Y, G140C, and E157Q [96,97], while other IN mutants, some of which include INSTI-resistant double and triple mutants, were identified in single round infection assays performed in cultured cells (Table 2). Phase 2 clinical trials suggested that resistant viruses can emerge within two weeks of the initiation of an EVG-based therapy [98,99]. In addition to the findings obtained in the in vitro experiments, which suggested that EVG has a resistance profile that is similar to RAL, clinical data showed that HIV-1 strains obtained from patients failing an RAL-based therapy displayed a decreased susceptibility to EVG [97]. Furthermore, a number of complex IN mutants (“complex” is defined here as an IN mutant with three or more amino acid substitutions) showed large decreases in susceptibility to EVG in single round replication assays. These data suggest that EVG tends to lose potency as the number of IN mutations increases [15]. Although EVG has the advantage that it can be used in once daily dosing regimens [90], it appears likely that, as the population of INSTI-experienced patients increases, and there are more virological failures, neither RAL nor EVG will be useful in salvage therapies.

The structure of EVG bound to the active site of the PFV intasome showed that the binding of EVG is similar to the binding of RAL [28]. Like RAL, EVG has a metal chelating motif that binds the two Mg^2+^ ions in the active site and a halobenzyl moiety that hydrophobically stacks with the base of the penultimate cytosine near the 3′ end of the viral DNA end. The isopropyl substituent of the hydroxyethyl moiety on the quinolone scaffold makes a Van der Waals interaction with PFV IN P214 (corresponds to HIV-1 IN P145) (Figure 3).

The similarities in the binding of RAL and EVG explain why the same mutations cause both compounds to lose potency. As mentioned above, the antiviral potencies of both RAL and EVG are severely compromised by the IN-resistant mutant G140S/Q148H [75]. Moreover, single round infection assays have shown that a large number of INSTI-resistant complex mutants that include G140S/Q148H can cause significant reduction in potencies for both RAL and EVG [15]. The most likely explanation for the effects of these mutations lies in the structure of the INSTIs. It appears that both RAL and EVG are not able to adjust their binding mode when challenged by changes in the active site of the mutant INs. If an INSTI cannot conform to changes in the binding pocket, mutations in the active site of HIV IN, such as G140S/Q148H, can affect the binding affinity of the drug. There can also be problems if a drug has functional groups that protrude from their centralized pharmacophores. For example, the oxadiazole group of RAL that hydrophobically stacks with Y143 in HIV IN specifically selects for resistance mutations at position 143. Generally speaking, having modifications that project beyond the substrate envelope permits the development of mutations that cause a steric clash with the projecting modification, reducing the binding of the drug (discussed below).

## 4. Second Generation INSTIs

### 4.1. Dolutegravir (DTG)

DTG was produced by a collaboration between Shionogi (Osaka, Japan) and Glaxo-SmithKline (GSK, Brentford, England) [23]. DTG was based on monocyclic carbamoyl pyridine derivatives that displayed promising antiviral activities and favorable pharmacokinetic profiles [100]. However, the resistance profiles of compounds that were based on this scaffold were poor. Optimization resulted in the development of a series of promising bicyclic compounds [101]. The addition of a hydroxyl group to the piperazinone ring resulted in a considerable increase in potency against the troublesome Q148K mutant (the Q148K mutant was used in the initial experiments; it behaves similarly to the Q148H mutant that is more frequently seen in the clinic). The hydroxyl was converted into tricyclic 5- and 6-membered ring analogs to avoid problems with the stability of the compounds [56]. Substitutions at the 4-position of the 6-membered ring and substitutions at the 3-position on the 5-membered ring resulted in the development of two compounds, DTG and CAB [23,56,58]. DTG, in combination with two NRTIs, was approved by the FDA in 2013 and is currently used in the most frequently recommended combination therapies for HIV-1 infections (Appendix A) [2]. Recently, a two-drug DTG/lamivudine (3TC) regimen has been added to the recommended list (for use HIV-1 patients whose HIV-1 RNA levels are below 500,000 copies per mL) [102,103,104]. In in vitro single round replication assays, DTG potently inhibited the replication of WT HIV-1 and nearly all of the IN-resistant mutants that were selected by RAL and EVG [23,29,74,75]. Importantly, DTG retained considerable potency against the well-known RAL-resistant IN mutants Y143R, N155H, and G140S/Q148H and the EVG-resistant mutants T66I and E92Q. The clinical efficacy of DTG is supported by the SPRING-1 and SPRING-2 clinical studies [21,105,106]. Approximately 90% of patients in both trials who received DTG as part of their therapy saw their viral RNA levels fall below the 50 copies per mL threshold.

When compared with resistance to first-generation INSTIs, resistance to DTG has been much more modest [19,20,21,22]. In vitro selection studies have produced relatively few DTG-resistant IN mutants [107]. The IN mutant that was most commonly selected by DTG was R263K [108]. Additional mutants have been reported to be selected by DTG, including H51Y, T66A/I, G118R, E138K, and S153Y/F [23,108,109,110]. Although DTG resistance is not readily selected in treatment-naïve patients in combination ART regimens, there has been a report that in a treatment-naïve patient with HIV-1 subtype F, DTG selected for both G118R and R263K [111]. There have been reports that DTG-resistant viruses can be selected in treatment-experienced, INSTI-naïve subjects. The most commonly selected IN mutant is R263K [112], followed by G118R [113,114]. Other IN mutants that were less frequently seen include: H51Y, T66I, E138K/T, Q148K, N155H, and S230R [112,113,114,115,116]. Resistant strains were selected in patients on DTG monotherapy. N155H was selected several times [117,118,119]; G118R, S147G, Q148H/R, and R263K were also found, although less frequently [110,117,118,120,121]. Some patients who failed a first-generation INSTI-based therapy (RAL or EVG) and were switched to a salvage therapy that included DTG had viral loads that were not well suppressed. The most common DTG-resistant IN mutants were L74I/M, T97A, E138A/K/T, S147G, N155H, and G140S/Q148H [16,17]. Several of the selected IN mutants were complex, having multiple amino acid substitutions at positions in and around the IN active site. Many of these complex IN mutants included the well-known G140S and Q148H mutations and had additional mutations at secondary positions, for example, T97A. However, generally speaking, the magnitude of the drop in susceptibility to DTG for these mutants was smaller than the drop in susceptibility for RAL and EVG. As a consequence, twice-daily doses of DTG can effectively suppress the replication of some IN mutants [17].

In in vitro susceptibility assays, numerous IN mutants have been identified that have decreased susceptibility to DTG (Table 3) [15,23,122]. Although it is not clear whether all of these IN mutants will appear in the clinic, the available data, taken together, show that there are multiple ways for HIV-1 to develop resistance to DTG. As is always the case, the resistant mutants are less able to replicate than WT [15,24,123]. The single round infectivities of some of the IN mutants described in Table 2 are much lower than WT HIV-1; however, seventeen of the twenty-five mutants are between ~40 to ~65% of WT HIV-1, and one IN mutant is near 70% of WT HIV-1 [15,24]. The resistance data, taken together with the replication data, show that there is a real possibility that DTG-resistant mutants will be seen more frequently in patients in the future, and that the problem is likely to be worse in patients who have failed a therapy that included a first-generation INSTI.

The initial structural studies that showed how DTG binds to the active site of IN were performed using the PFV intasome [29]. The binding of DTG to PFV IN is quite similar to the binding of RAL. DTG chelates the Mg^2+^ ions in the active site, and the halobenzyl group hydrophobically π-π stacks with the base of the penultimate cytosine, displacing the terminal adenine (Figure 4). However, there are important differences in the structures of the two compounds. In DTG, the linker that joins the halobenzyl moiety and tricyclic scaffold is both longer and more flexible (Figure 1 and Figure 4, red circles). The longer linker allows a bound DTG to adjust to the changes in the active site of IN mutants, retaining the ability to chelate the Mg^2+^ ions and stack the halobenzyl with the penultimate cytosine base on the viral DNA. Understanding the importance of the length of the linker has facilitated INSTI development; a longer linker has been a structural feature of recently developed INSTIs [25,124,125,126]. In addition, the tricyclic scaffold of DTG stacks with the base of the displaced terminal adenine, increasing its association within the IN active site [127].

Structural analyses of DTG bound to the active SIV_rcm_ IN have pointed to the importance of the oxazine ring (the “A” ring, which is the ring system farthest from the halobenzyl group; Appendix A, black circle) [26]. The oxazine ring makes close contacts with N117 and G118 of the α2β4 loop. Although HIV-1 does not readily develop resistance to DTG, at least in treatment-naïve patients, some clinical studies have reported adverse effects. Many of the reported adverse events were neurological, and there are reports in treatment-experienced patients of an increase in neuropsychiatric disorders [128]. There are also reports of neural tube defects in children who were exposed to DTG at the time of conception [129]. However, because of its effectiveness and once-daily dosing, DTG has replaced EFV as the first-line treatment strategy for HIV-1 infections in South Africa [130].

### 4.2. Bictegravir (BIC)

BIC, which is the most recent FDA-approved INSTI, was developed by Gilead Sciences [25]. BIC is similar in structure to DTG. The key difference between BIC and DTG is in the “A” ring. BIC was designed and developed from DTG using structure activity relationship (SAR) data by first modifying the “A” ring with a series of diaza-bridging bicycles (Appendix A). However, the initial changes that were made did not improve the antiviral potency against the IN double mutant G140S/Q148R (like Q148K, the Q148R mutant is expected to behave similarly to Q148H). A different “A” ring modification that used oxaza-bridging led to an improvement in potency against G140S/Q148R. Using this “A” ring modification as the centralized pharmacophore, SAR studies were performed on the benzyl moiety, which showed that a 2,4,6-trifluorophenyl gave the best solubility and potency against G140S/Q148R. This, in turn, led to the development of BIC [131]. BIC potently inhibited the replication of WT HIV-1, the first-generation INSTI-resistant mutants, and, more importantly, retained some of its potency against G140S/Q148H IN and some of the DTG-resistant mutants [15,25,75]. Clinical trials confirmed the in vitro findings. In multiple phase III studies with treatment-naïve participants, BIC in combination therapies showed non-inferiority to DTG [132,133,134]. A smaller percentage of the participants reported adverse effects when taking BIC compared to DTG. However, therapies that include BIC have been associated with weight gain (Appendix A) [135].

There are, to date, no reports that resistance to BIC has developed in patients [18]. IN mutants that reduce the potency of BIC have been identified in vitro in which BIC primarily selected for the M50I, S153F/Y, R263K, and M50I/R263K IN mutants [25,136]. Other mutants selected by either DTG or CAB that cause a reduced susceptibility to BIC in single round replication assays are shown in Table 4 [15,24].

There are recent structures of BIC bound in the active sites of the HIV-1 and SIV_rcm_ intasomes [26,27]. BIC binds similarly to RAL, EVG, and DTG; the chelating motif of BIC coordinates the Mg^2+^ ions and the halobenzyl moiety stacks with the base of the penultimate cytosine. The displaced 3′-adenosine can adopt several rotameric conformations. The oxazepine (“A”) ring of BIC (Appendix A), which seems to be a critical element of this INSTI, makes several contacts with the β4α2 loop near the IN active site (Figure 5, panel A). The methylene bridge in the oxazepine ring system may impart some rigidity that helps BIC retain the ability to bind in spite of the changes in the geometry of the IN active site when there are amino acid substitutions in the β4α2 loop. Currently, BIC is recommended for treating both treatment-naïve and treatment-experienced patients, and BIC-containing ART regimens are some of the best currently available therapeutic options (Appendix A) [2].

### 4.3. Cabotegravir (CAB)

CAB was produced by a collaboration between Shionogi and GSK and was recently approved for use in Canada and now in the United States [59,60]. It is structurally similar to DTG and BIC; the centralized pharmacophore is a tri-cyclic ring with the characteristic “di-keto” metal ion chelating group with the same extended linker connecting the core to a halobenzyl group. As discussed earlier, the “A” rings of CAB, DTG, and BIC are different. In CAB, the “A” ring is a 5-membered oxazole, which is smaller than the six-membered oxazine ring of DTG and seven-membered oxazepine ring of BIC (Appendix A). CAB is relatively insoluble in aqueous media and, probably as a consequence, has a long half-life in vivo; the half-life of CAB is close to 40 days, as opposed to 12 h for DTG [137]. Although CAB is relatively insoluble, it can be formulated in a nanoparticle suspension and used as a long-acting antiretroviral (see below). In the long-acting antiretroviral treatment enabling clinical trial (LATTE-1), oral CAB plus two NRTIs suppressed viral replication for 24 weeks [58], and there was continued suppression of the virus in subjects who were switched to a dual therapy that consisted of CAB and the NNRTI rilpivirine (RPV) for 96 weeks [7]. The results were similar to the DTG versus EFV trials, establishing CAB as a clinically relevant INSTI. In single round in vitro replication assays, the potency of CAB against HIV-1 is high [15,59]; however, there is a considerable loss of potency against the well-known INSTI-resistant mutant G140S/Q148H (FC = 15) [15]. Single round replication assays have shown that CAB loses considerable potency against the DTG-resistant mutants G118R (FC = 8), R263K (FC = 7), and H51Y/R263K (FC = 10) [15]. Several IN mutants were selected in vitro using CAB [59,138]. When in vitro selection studies were initiated with any of the IN Q148H/K/R single mutants or clinical isolates, additional mutations in IN were selected. IN mutants were also identified with in vitro replication assays. Several first-generation INSTI-resistant viruses displayed decreases in susceptibility to CAB (Table 5) [15,24]. Modeling CAB onto BIC in the active site of the HIV-1 intasome suggests that the mode of binding is similar, involving chelation of the Mg^2+^ ions and hydrophobic stacking of the benzyl moiety with the penultimate cytosine base near the 3′ end of the viral DNA end (Figure 6). Although the resistance profile of CAB is inferior to DTG and BIC, the fact that CAB can be used in long-acting formulations suggests that it has important but specific clinical potential (discussed below).

## 5. Using Structural Analyses to Understand the Mechanism(s) of INSTI Resistance

The available structures of the INSTIs in the active sites of HIV-1, SIV_rcm_, and PFV intasomes can be used as models to propose explanations for the effects of INSTI-resistant mutations on the susceptibilities of the second-generation INSTIs [26,27]. Those who are interested in this problem can consult the two recent reviews that are focused on how changes in the structures of HIV-1/SIV_rcm_ intasomes caused by mutations in IN affect the potencies of the INSTIs [139,140]. The mutations in the drug-resistant viral variants found in populations of treated individuals can be mapped onto the active sites of IN. However, it is intasomes, not free IN, that are targeted by INSTIs, and intasomes should be used in these analyses. As described above for individual drugs, the base of the penultimate cytosine near the 3′ end of the viral DNA makes a π-π interaction with the halogenated benzene moiety of the INSTI. In turn, the displaced adenine at the very 3′ end of the viral DNA can stack on top of the core of the INSTI scaffold, although an alternative non-interacting conformation was also seen. To date, both the structures of PFV and SIV intasomes have been used to study the binding of first- and second-generation INSTIs and the mechanisms that underlie resistance [26,27]. We now have structures of WT HIV intasomes with and without a variety of bound INSTIs, and structures of drug-resistant forms of HIV intasomes should become available soon. Models comparing the active sites of HIV-1, SIVrcm, and PFV intasomes with either DTG or BIC bound in the active site highlight some of the important resistance mutations (Figure 7). As noted in the legend, positions at which the amino acid is not conserved with HIV are starred. In SIV_rcm_ IN, these include I74, T138, V151, and G230, which are L74, E138, I151, and S230 in HIV-1 IN, respectively. In PFV IN, these include P111, I112, I130, P161, S209, S217, V220, R222, D226, A328, and D367, which are M50, H51, T66, E92, G140, Q148, I151, S153, E157, S230, and R263, respectively, using a structure-based alignment (due to differences in the length of the linker connecting individual domains, a sequence-based alignment should not be used to compare these variants). From this analysis, it is clear that while the overall binding mode of the compounds is similar for all three intasomes, there are important differences in specific residues in or near the active sites (this is especially true for the PFV intasome). Thus, it is important to use the lentiviral intasomes for analyzing mechanisms of drug resistance. The problem becomes particularly acute when complex mutants are analyzed and/or if the structural insights are used for drug design, which is why the new HIV intasome structures are particularly important.

Because the G140/Q148 mutants are broadly associated with resistance to the available INSTIs, we will briefly consider what is believed to be the underlying mechanism. There is an important interaction between a particular bound water molecule, behind the active site, the catalytic residues D116 and E152, and the nearby residue Q148 [26,27,140]. If the amino acid at position 148 is replaced (Q148H/K/R), the bound water molecule is expelled, which means that this key interaction is also lost. The Q148H/K/R substitutions increase the local electropositive charge, which in turn affects the electron density in the Mg^2+^ -ligand cluster. This effect is increased by hydrogen bonding interactions that involve G140S and Q148H. These new electropositive charges reduce the strength of the interaction between the chelating motif of the INSTI and the Mg^2+^ ions in the IN active site [26,140]. Secondary IN mutations that are associated with G140S and Q148H, including E138A/K, help stabilize a hydrogen bond network. These networks feed back to the amino acid at position 148, which, in turn, affects the binding of the INSTIs to the Mg^2+^ -ligand cluster [24]. The Q148H/K/R mutants also lead to a decrease in replication capacity. In turn, the G140S substitution compensates for this loss of viral fitness. Single round infection assays have shown that the second-generation INSTIs lose efficacy against complex IN mutants that comprise G140S, Q148H, and one or two additional amino acid substitutions at other IN positions (C56, V72, L74, V75, T97, T122, E138, G149, and G163). If efforts to make an INSTI that is broadly effective against a wide variety of resistant variants, including those that involve G140S and Q148H, are not successful, it might be necessary to develop compounds that are specifically targeted against this class of mutants. The new compounds could be used in combination with a standard INSTI, such as BIC, which would target WT HIV-1 and most, or all, of the non-G140/Q148 INSTI-resistant mutants. This type of dual therapy could potentially be used in treatment-experienced, INSTI-experienced patients who have virological failure.

Additional modeling suggests that the effects of the G118R mutant on the susceptibility of IN to INSTIs are based on the close contacts between the “A” rings of DTG and BIC with the β4α2 loop of HIV-1 IN (Figure 5, panel A) [26]. The G118R mutation could perturb these interactions and affect the binding of INSTI to the intasome active site (Figure 5, panel B). Modeling the effects on the susceptibility of R263K to DTG and CAB is difficult because of the multimeric nature of IN, which was mentioned previously [24]. HIV IN is (mostly) tetrameric in solution; however, the intasome that inserts the ends of HIV DNA into the host genome in infected cells could comprise as many as sixteen IN monomers [141]. If there is substitution at position R263, this substitution will be present in all of the subunits in the intasome. Because the various protomers play different roles in the intasome, a single amino acid substitution can have different effects on the different IN protomers. In the case of the R263K mutation, the effects could include disrupting the interaction of the guanine nucleotide of the non-transferred strand with the penultimate cytosine of the transferred viral DNA strand, which could, in turn, affect the positioning of the end of the viral DNA and the hydrophobic stacking interaction between the benzyl moiety of the INSTIs and the base of the penultimate cytosine. The R263K mutant could also prevent the interaction of Q146 with halogen atoms on the benzyl moiety of an INSTI, which supports the idea that R263K could affect the positioning of the viral DNA in the active site of IN. There are other interactions that could affect the positioning of the viral DNA strand [24]. Structures of some of the most important HIV-1 IN mutants in complex with some of the most potent INSTIs should be forthcoming. These structures could help to explain the mechanisms of resistance and provide guidance for the development of new INSTIs that should be able to retain efficacy against additional INSTI-resistant mutants.

## 6. Replication Capacity of the IN Mutants

The ability of the mutant virus to replicate plays an important role in the development of resistant strains. In single-round assays, the infectivity of the primary RAL mutants is (relative to WT HIV-1): Y143R (~40%), N155H (~50%), and Q148H (~50%). Although the single round infectivity of an IN mutant can easily be determined in vitro and compared to WT HIV-1 [15,24], this measurement does not always correlate with the behavior of the mutant virus in the patient. There are some clinically relevant IN mutants that show larger reductions in single round infectivity assays than mutants that are not seen as often in the clinic. For example, an amino acid substitution at position Q148 has a greater impact on infectivity than substitutions at T66, L74, or T97 [15], and the single-round infectivity of Q148H (50% of WT) is lower than L74M, which is 85–90% of WT. There are other clinically relevant IN mutants that have larger effects on viral replication in a single round assay: G118R (23%), Q148H (52%), and Q148K (15%). If a mutant virus is resistant, but replicates poorly, there will be a selection for additional substitutions that improve the ability of the resistant virus to replicate; such changes are commonly referred to as compensatory mutations [142,143]. When G140S was added to Q148H, replication was improved in a single round infectivity assay, to ~65%, suggesting that the IN double mutant G140S/Q148H has a replication advantage relative to the Q148H mutant [83]. There is also an increase in replication when a E138K amino acid substitution is added to Q148K from 14 to 50% of WT.

Unfortunately, most INSTIs lose some, and some INSTIs lose considerable potency against the double mutants E138K/Q148K, G140S/Q148H, and G140S/Q148K [15]. There are complex mutants, based on these double mutants, which have higher single round infectivities that help the mutant IN evade the available INSTIs. Although the replication capacity of INSTI-resistant mutants is important, for most of the known complex IN mutants, increasing the number of secondary mutations does not have a large effect on single round infectivity. The effects of the additional mutations appear to be primarily on the susceptibility of the virus to the INSTI [15,24]. This suggests that when the virus is confronted with a potent INSTI, there is considerable selective pressure on the virus to develop a high level of resistance to the drug. That result is perhaps related to the fact that in an infection, IN must carry out a total of four enzymatic steps (two 3′-processing reactions and two strand transfer reactions). Thus, the virus can tolerate some reduction in the activity of IN; however, if an INSTI is able to block the integration reaction, it is fatal for the virus.

## 7. Novel Mechanisms of INSTI Resistance

What appear to be two novel mechanisms of resistance to INSTIs have been reported in recent years. In a population of individuals who were infected with HIV-2 and treated with RAL, a mutant virus was found that has five amino acids inserted into the C-terminal domain (CTD) of IN. This insertion mutation conferred resistance to RAL. Importantly, these viral isolates were also moderately resistant to both DTG and CAB in vitro [144]. Because the CTD makes a critical contribution to the active site where INSTIs bind, insertion of amino acids into the CTD may reduce the ability of the INSTIs to bind to the HIV-2 intasome. Whether an analogous mutant could arise in HIV-1 IN is not known. An even more unusual set of INSTI-resistant mutants have been reported, by multiple groups, in the 3′-polypurine tract (3′PPT) of the viral genome. In these mutants, the GGGGGG sequence normally present in the 3′PPT was mutated to GGGAGT, GGGGAGT, GCAGT, GGAGTG, or GGGAGC [119,145]. The mutant viruses were found in both individuals infected with HIV [17] and in cells that were infected in vitro and were treated with DTG [145]. Mutations in the PPT can affect the specificity of RNase H cleavage at the 3′-end of the PPT during reverse transcription [146]. If the site where RNase H cleaves the genome is perturbed, it would change the sequence at the end of the 3′ long terminal repeat (LTR). Changes in the sequence at the end of the viral DNA could have important implications for both viral replication and for DTG resistance. First, as has already been discussed, the viral LTRs must form a complex with HIV IN in which the 3′ end of the viral DNA can be processed by IN to generate the substrate for the strand transfer reaction. Then, the processed ends of the viral DNA and the host DNA need to bind to the active site of IN so that the viral DNA is inserted into the host genome. For an INSTI to inhibit the strand transfer reaction, the terminal CA dinucleotide, found at the 3′ end of the processed viral DNA, must be displaced and physically interact with the bound INSTI. Changes in the sequences at the end of the 3′ LTR could affect the structure or the enzymatic activities of an assembled intasome or the assembly of the intasome itself, both of which could also affect the binding of DTG. Even if we assume that the mutations that were reported can confer resistance to DTG, the mutation in the PPT would only affect one of the two viral DNA ends. Presumably, the IN-mediated integration of the second DNA end (the 5′ LTR end) would still be blocked by DTG. However, there is evidence, from experiments performed on cultured cells, that if one viral DNA end is inserted by IN, the second end can be inserted, with reasonable efficiency, by host enzymes. This could lead to insertions, deletions, and translocations that are similar to those that have been reported for viruses in which one end of the viral DNA is mutated [147], or seen when INSTI treatment is suboptimal [148], and for some drug-resistant mutants [149]. Although PPT mutations may be rare in patients [150], the mechanism of resistance raises interesting questions and is the subject of ongoing investigations, at both the biochemical and structural level.

## 8. Long-Acting Formulas, Implants, and PrEP

Currently, most treatment-naïve patients are prescribed a three-drug combination ART regimen for HIV-1 infections (Appendix A). This strategy is highly effective, but resistance can arise in individuals who are not strictly compliant. In addition, there are concerns about the potential effects of life-long consumption of potent anti-HIV-1 drugs. There is a special concern that NRTIs, which are used as the “backbone” of most current treatments, can interfere with DNA replication in host cells. The mitochondrial DNA polymerase seems to be particularly susceptible to NRTIs. Mitochondrial toxicity has been associated with a higher incidence of peripheral neuropathy and with bone marrow suppression [41,151]. In addition, NRTIs have been implicated in cardiovascular disease; patients have been shown to have an increased risk of myocardial infarction [152].

There is precedent for using long-acting formulations to administer contraceptives and antipsychotics [153,154,155]. Although a shift from a daily oral dosing to long-acting ART formulations could help with compliance, long-acting formulations will not solve the issues associated with the toxicities of the drugs. Trials with long-acting ART formulations have given promising results, both in macaque models and in human trials [6,7,8,156,157]. A number of approaches, including formulating INSTIs into intravaginal rings, injectables, or implants are being tested for both therapeutic and prophylactic applications. CAB, which has a long half-life, is a prime candidate for use in long-acting formulations. Recent clinical trials (the first long-acting injectable regimen (FLAIR) and the antiretroviral therapy as long-acting suppression (ATLAS) phase 3 studies) suggested that patients who were injected monthly with a dual formulation (CAB and RPV) maintained viral suppression as well as patients on a standard three-drug daily oral formulation [156,157]. The use of CAB and RPV in combination avoids the potential problems that are associated with the long-term administration of an NRTI.

Another therapeutic option that is now being tested is to wait until the virus has been suppressed by one of the three-drug combination regimens and then switch to a maintenance treatment based on only two drugs, i.e., an INSTI with an NRTI or NNRTI. Such strategies include Juluca (DTG and RPV) [124], and more recently, Dovato, which is composed of DTG and the NRTI lamivudine (3TC; Appendix A) [5,158]. Both approaches, a long-acting injectable or a two-drug maintenance therapy, are expected to be well tolerated, which should improve adherence by patients.

CAB also has the potential for be used for long-acting PrEP. The HPTN 083 clinical trial showed that CAB (injected monthly) effectively prevented HIV-1 infection for gay and bisexual men and transgender women when compared to Truvada [12,159], which is composed of emtricitabine and tenofovir disoproxil fumarate (FTC/TDF) [11].

## 9. Future Perspectives

We now have INSTIs that are potent, have minimal toxicity, and are broadly effective against many of the known IN mutants. There are still problems with resistance, and resistance will probably increase as second-generation INSTIs are given to greater numbers of infected individuals. New INSTIs are being designed and developed; however, it is likely that any new INSTIs will be structurally related to the best of the FDA-approved INSTIs. For a new INSTI to afford substantial improvement, it would need to address at least some of the problems presented by the IN mutants G118R, R263K, and G140S/Q148H and the complex IN mutants that include these variants in combination with additional mutations [15,24]. Recent structural work using both HIV-1 and SIV intasomes (see these reviews for additional information about recent structural analyses [139,140]) has confirmed and extended the previous structural studies performed with PFV IN [28,29], and has already helped in the design of new INSTIs. It is likely that the structures of HIV-1 intasomes with a number of well-known IN mutants, with bound INSTIs, will be solved in the near future. Those structures should provide additional information that can be used in the design of better INSTIs.

We have known for several years that there are several important criteria that influence whether INSTIs based on the current scaffolds will be broadly effective against IN mutations. These include: (1) the chelating motif that binds the two Mg^2+^ ions at the active site should be coplanar; (2) the linker that connects the halogenated benzyl moiety and the centralized pharmacophore needs to be long enough to allow the INSTI to adapt to changes in the active site; (3) it appears, based on the structures of the most effective drugs, that having at least two halogen modifications on the benzyl ring that stacks with viral DNA is beneficial. Recent structural work performed with HIV-1 and SIV_rcm_ IN has identified IN residues (for example, residues in the α2β4 loop near the IN active site) that help determine what should be modified (and optimized) on the pharmacophore [26,27]. In addition, the substrate envelope, the space that is occupied by the DNA substrates (including the preprocessed viral DNA or target DNA) when they bind to IN, has been defined by structural analyses [27,125,126,160]. Defining the substrate envelope is an important concept, because identifying where the DNA substrates contact IN should help in the design of inhibitors that bind entirely within the substrate envelope. As has already been discussed, it is more difficult for a viral enzyme to develop resistance if the inhibitor binds within the substrate envelope. Thus, future INSTIs are likely to bind mostly or entirely within the substrate envelope, and the most successful INSTIs will probably be compounds that effectively fill the substrate envelope and contact the conserved residues that bind the substrates [125,126].

It may be possible to develop compounds that make better contacts with the invariant adenosine nucleotide located at the 3′ ends of the viral DNA. The adenine is a critical part of the CA dinucleotide found in all retroviral LTRs [161,162,163,164,165]. DNA mutations in this nucleotide have major consequences for viral replication [163,164,166]. In addition, INSTIs designed to interact more strongly with the terminal adenosine nucleotide could potentially better adapt to the changes in the geometry of the IN active site that accompany some of the most problematic IN mutations.

## 10. Conclusions

INSTIs, in particular second-generation INSTIs, have significantly improved the combination ART therapies that are available to HIV-1 patients. In addition, INSTIs hold great promise for PrEP. It appears that, in the near future, patients will have broad access to long-acting formulations that will help with adherence, both for therapy and for PrEP. It also appears that increased use of INSTIs could decrease the risk of toxicities that have been associated with long-term ART. Given the considerable success that has been achieved with the available INSTIs, the development of new INSTIs should focus on either developing compounds for the emerging approaches (long-term/slow-release formulations and/or PrEP), overcoming the most problematic of the known mutants, or both. Thus, early in their development, promising new compounds should, at a minimum, be screened against IN mutants that are known to reduce the susceptibility of HIV IN to some of the second-generation INSTIs, for example, G118R, N155H, R263K, and G140S/Q148H. These IN mutants (and more complex mutants that include these mutations) cause decreases in susceptibility to some of the best available INSTIs. Compounds that retain potency against the initial set of mutants could then be tested against more complex IN mutants and for their ability to select for additional mutants in cell culture. Fortunately, recent structural analyses of the second-generation INSTIs bound to HIV-1 and SIV_rcm_ intasomes have already provided information that should be useful in informing the design of new INSTIs. Solving structures of the most problematic mutants should facilitate the design and development of INSTIs that are more broadly effective against the known mutants.

## Figures and Tables

**Figure 1 viruses-13-00205-f001:**
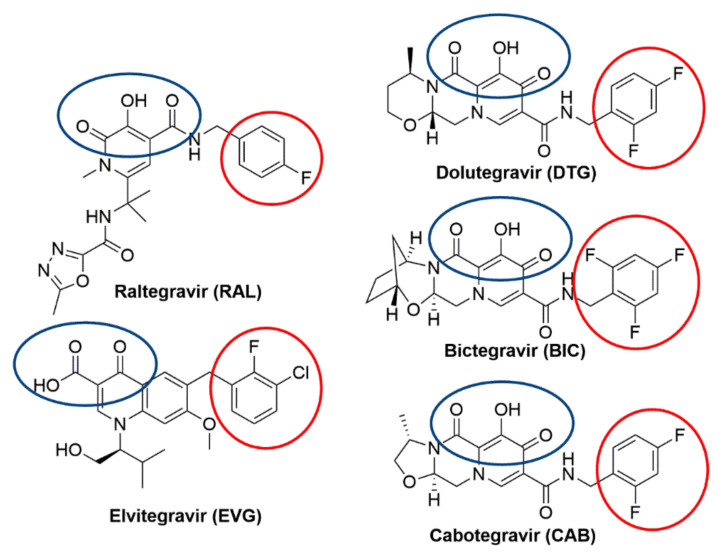
Chemical structures of the clinically relevant integrase strand transfer inhibitors (INSTIs). Chemical structures of the clinically relevant INSTIs are shown. The chelating motifs on the centralized pharmacophores that interact with Mg^2+^ cofactors in the integrase (IN) active site are highlighted with a blue circle. The halobenzyl moieties, which are connected to the centralized pharmacophore by a linker group, are circled in red.

**Figure 2 viruses-13-00205-f002:**
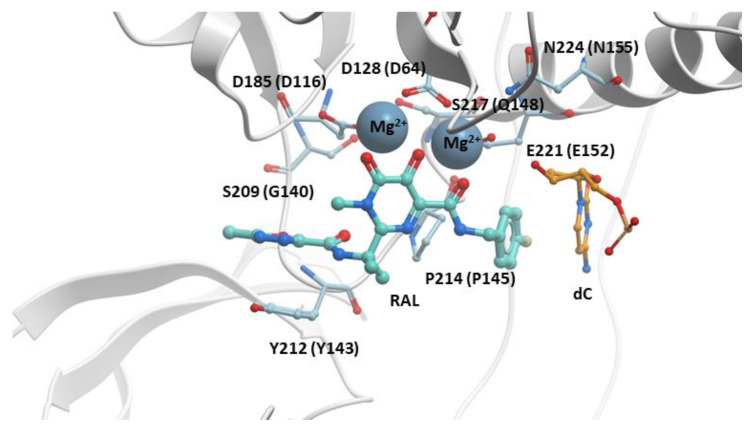
RAL in the active site of the PFV intasome. RAL (cyan) is shown chelating the Mg^2+^ cofactors in the active site of the PFV intasome. PFV IN is in white, and the penultimate cytosine near the 3′ end of the viral DNA is in orange; the adenine at the 3′ end of the viral DNA is omitted for clarity. PFV IN active residues D128, D185, and E221 are labeled (the corresponding HIV-1 IN residues are shown in parentheses). Additional IN residues that undergo resistance mutations are labeled with their corresponding HIV-1 IN residues given in parentheses.

**Figure 3 viruses-13-00205-f003:**
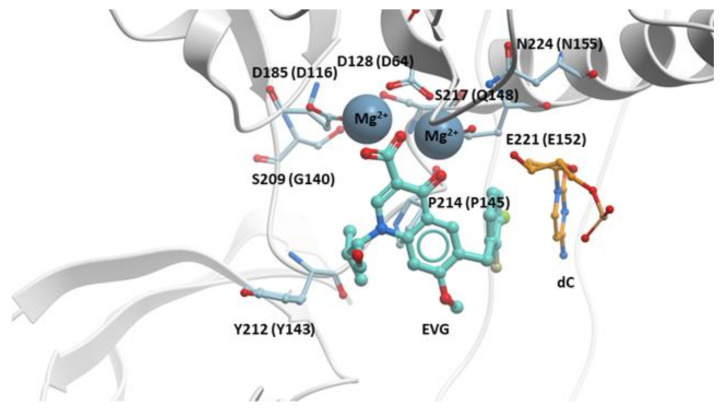
EVG in the active site of the PFV intasome. EVG (cyan) binding in the active site of the PFV intasome, chelating the Mg^2+^ cofactors (gray). PFV IN is in white, and the penultimate cytosine near the 3′ end of the viral DNA is in orange; the adenine on the 3′ end of the viral DNA is omitted for clarity. PFV IN active residues D128, D185, and E221 are labeled (corresponding HIV-1 IN residues are given in parentheses). Additional IN residues that undergo resistance mutations are labeled with the corresponding HIV-1 IN residues given in parentheses.

**Figure 4 viruses-13-00205-f004:**
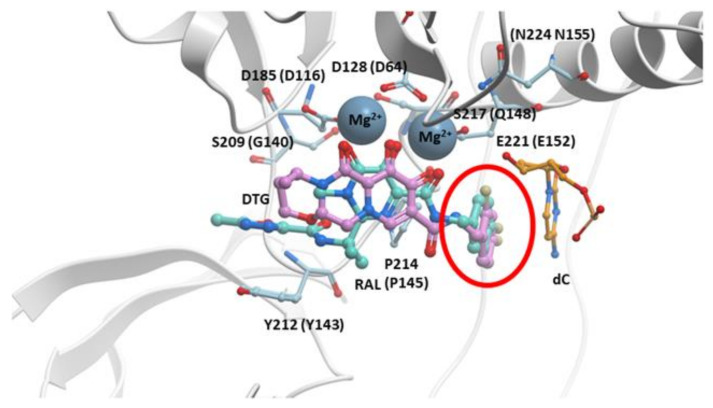
RAL overlaid onto DTG in the active site of the PFV intasome. The structures of both RAL (cyan) and DTG (purple) bound in the active site of the PFV intasomes [29]. IN active site residues D128, D185, and E221 are shown (the corresponding HIV-1 IN residues are given in parentheses). The penultimate cytosine near the 3′ end of the viral DNA end is labeled and colored orange as is IN residue Y212 (shown light blue; this residue is Y143 in HIV-1 IN) where mutations conferring resistance commonly arise. The halogenated benzyl moieties are highlighted by a red circle. The terminal adenine on the 3′ end of the viral DNA is omitted for clarity.

**Figure 5 viruses-13-00205-f005:**
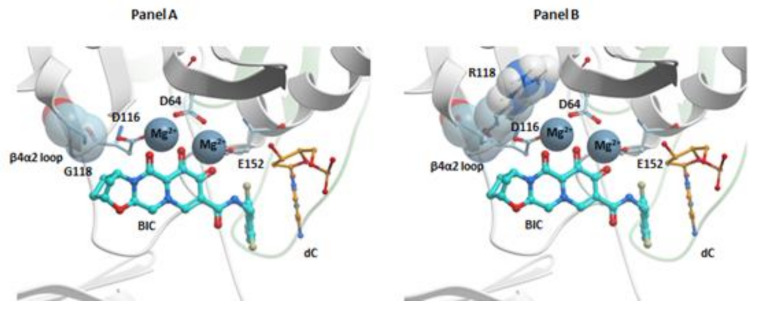
BIC in the active site of the HIV-1 intasome. Panel (**A**): BIC (cyan) is bound in the active site in the HIV-1 intasome. G118 of the β4α2 loop (labeled) is highlighted using a space filled model to reveal the close contact with the “A” ring of BIC. Panel (**B**): BIC (cyan) is shown in the HIV-1 intasome. The mutated amino acid, R118, is shown in a space filling model which shows how this amino acid substitution could disrupt the binding of BIC in the active site of the HIV-1 intasome. In both panels, the catalytic residues are shown and labeled. The penultimate cytosine is labeled and shown in orange, and the Mg^2+^ cofactors are gray. The terminal adenine on the 3′ end of the viral DNA end is omitted for clarity.

**Figure 6 viruses-13-00205-f006:**
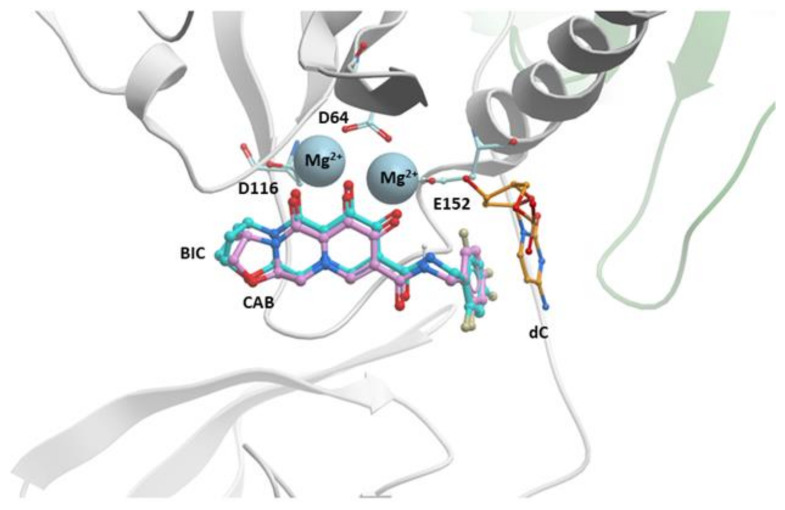
Modeling CAB onto the structure of BIC in the active site of the HIV-1 intasome. CAB (purple) is docked onto the structure of BIC (cyan) in the active site of the HIV-1 intasome. HIV-1 IN is shown; its catalytic residues are shown and labeled. The penultimate cytosine is shown (labeled), as are the Mg^2+^ cofactors (shown in gray and labeled). The terminal adenine on the viral DNA end is omitted for clarity.

**Figure 7 viruses-13-00205-f007:**
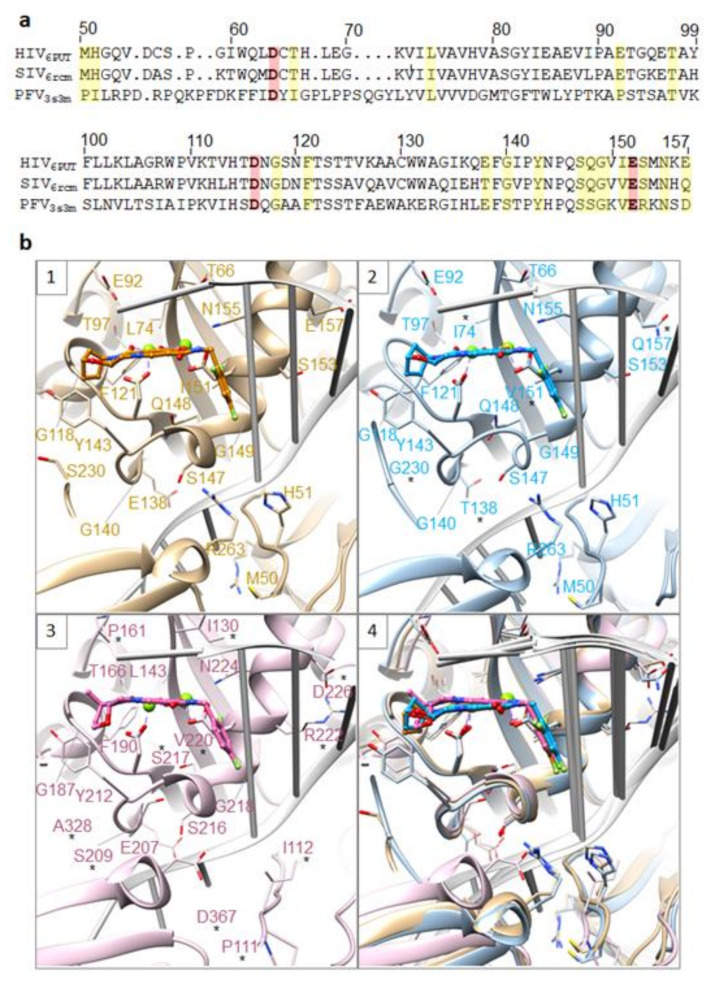
Comparison of HIV-1, SIV, and PFV intasomes. (**a**) Structure-guided alignment of the INs from HIV-1 (PDB ID: 6PUT), SIV_rcm_ (6rcm), and PFV (3s3m), spanning HIV-1 IN residues M50-E157. Yellow stripes depict HIV IN residues, and the corresponding homologs in SIV and PFV, that are involved in drug resistance. Red stripes highlight the catalytic DDE triad responsible for IN enzymatic activity. (**b**) Comparison of the catalytic cores of HIV-1 IN (6put, first panel in tan), SIV_rcm_ (6rcm, second panel in blue), and PFV (3s3m, third panel in pink) showing 20 residues that are involved in HIV drug resistance. All three structures are superimposed in the fourth panel. In the second and third panel, residues that are not conserved when HIV-1 IN is compared with either SIV_rcm_ IN (second panel) or PFV IN (third panel) are labeled with a star.

**Table 1 viruses-13-00205-t001:** Identification of raltegravir (RAL)-resistant mutants. IN single, double, and triple mutants that have been selected against RAL in patients, in vitro selection studies, or IN susceptibility assays are listed.

IN Single Mutants	IN Double Mutants	IN Triple Mutants
M50I	T66I/E157Q	T66I/T97A/E157Q
L74M	E92Q/N155H	L74M/G140A/Q148R
T97A	E138A/Q148R	L74M/G140C/Q148R
S119R	E138K/Q148K	E92Q/N155H/G163R
E138K	E138K/Q148R	T97A/G140S/Q148H
G140S	G140A/Q14H	T97A/Y143R/Q148H
Y143C	G140A/Q148K	T97A/Y143R/N155H
Y143H	G140A/Q148R	T97A/Q148H/N155H
Y143R	G140C/Q148R	E138A/G140S/Q148H
Q146L	G140S/Q148H	E138A/S147G/Q148R
Q146P	G140S/Q148K	E138K/G140A/Q148K
Q148H	G140S/Q148R	E138K/G140C/Q148R
Q148K	G140S/N155H	E138K/G140S/Q148H
Q148R	Y143R/Q148H	G140S/Y143R/Q148H
V151I	Y143H/N155H	G140S/Q148H/N155H
S153Y	Y143R/N155H	G140S/Q148H/G163K
N155H	Q148H/N155H	
G163R	Q148R/N155H	
	N155H/G163R	

**Table 2 viruses-13-00205-t002:** Identification of elvitegravir (EVG)-resistant mutants. IN single, double, and triple mutants that have been selected against RAL in patients, in vitro selection studies, or IN susceptibility assays are listed.

IN Single Mutants	IN Double Mutants	IN Triple Mutants
H51Y	H51Y/R263K	T66I/T97A/E157Q
T66I	T66I/E157Q	L74M/G140A/Q148R
E92Q	E92Q/N155H	L74M/G140C/Q148R
F121Y	E138A/Q148R	E92Q/N155H/G163R
G140C	E138K/Q148K	T97A/G140S/Q148H
S147G	E138K/Q148R	T97A/Y143R/Q148H
Q146L	G140A/Q148H	T97A/Y143R/N155H
Q148K	G140A/Q148K	T97A/Q148H/N155H
Q148R	G140A/Q148R	E138A/G140S/Q148H
S153Y	G140C/Q148R	E138A/S147G/Q148R
N155H	G140S/Q148H	E138K/G140A/Q148K
E157Q	G140S/Q148K	E138K/G140C/Q148R
R263K	G140S/Q148R	E138K/G140S/Q148H
	G140S/N155H	G140S/Y143R/Q148H
	Y143H/N155H	G140S/Q148H/N155H
	Y143R/N155H	G140S/Q148H/G163K
	Q148H/N155H	
	Q148R/N155H	
	N155H/G163R	

**Table 3 viruses-13-00205-t003:** IN mutants selected by dolutegravir (DTG). Single, double, and triple IN mutants that have been selected by DTG in patients, in vitro selection studies, or IN susceptibility assays are listed. The FC, the potency of DTG against the IN mutant relative to its potency versus WT HIV-1 as determined by single round replication assay, is also shown as are the single round infectivity of the IN mutant relative to WT HIV-1.

IN Mutant	FC (Potency Against Mutant HIV-1 IN Relative to WT Potency)	Single Round Infectivity (Relative to WT HIV-1)
G118R	8	23.0 ± 2.2
R263K	7	58.5 ± 10.5
H51Y/R263K	10	13.8 ± 3.9
L74M/Q148R	7	42.3 ± 6.9
E138K/Q148K	16	49.4 ± 7.5
G140A/Q148K	282	12.3 ± 1.3
G140S/Q148R	16	32.6 ± 7.0
M50I/S119R/R263K	28	61.4 ± 7.8
C56S/G140S/Q148H	7	69.3 ± 11.0
V72I/E138K/Q148K	68	47.4 ± 4.3
L74M/G140A/Q148R	8	32.5 ± 4.4
L74M/G140C/Q148R	6	29.4 ± 7.3
V75A/G140S/Q148H	11	56.2 ± 6.1
T97A/G140S/Q148H	35	55.9 ± 13.1
T122N/G140S/Q148H	37	65.0 ± 5.3
E138K/G140A/Q148K	133	42.9 ± 7.5
E138K/G140S/Q148H	43	54.4 ± 15.9
G140S/Q148H/G149A	67	62.0 ± 8.8
G140S/Q148H/N155H	49	63.0 ± 14.9
G140S/Q148H/G163K	15	49.6 ± 17.2
C56S/G140S/Q148H/G149A	28	65.1 ± 7.8
L74M/V75A/G140S/Q148H	55	53.4 ± 7.5
L74M/E138K/Q148R/R263K	6	13.1 ± 3.2
L74M/G140S/S147G/Q148K	221	23.2 ± 4.5
T66I/L74M/E138K/S147G/Q148R/S230N	33	61.6 ± 15.1

**Table 4 viruses-13-00205-t004:** IN mutants selected against bictegravir (BIC). IN double and complex mutants that have been selected against BIC using in vitro selection studies or identified from IN susceptibility assays are shown. The FCs of the potencies of BIC against the IN mutant relative to its potency against WT HIV-1 are also included as are the single round infectivity of the IN mutant relative to WT HIV-1.

IN Mutant	FC (Potency Against Mutant HIV-1 IN Relative to WT Potency)	Single Round Infectivity (Relative to WT HIV-1)
M50I/R263K	5	70.2 ± 6.1
S119R/R263K	6	61.1 ± 6.2
T124A/S153Y	5	N/A
E138K/Q148K	31	49.4 ± 7.5
G140A/Q148K	72	12.3 ± 1.3
G140A/Q148R	5	48.6 ± 8.1
M50I/S119R/R263K	7	61.4 ± 7.8
V72I/E138K/Q148K	36	47.4 ± 4.3
L74M/G140A/Q148R	6	32.5 ± 4.4
V75A/G140S/Q148H	5	56.2 ± 6.1
T97A/G140S/Q148H	16	55.9 ± 13.1
T122N/G140S/Q148H	8	65.0 ± 5.3
E138K/G140A/Q148K	117	42.9 ± 7.5
G140S/Y143R/Q148H	5	52.3 ± 1.1
G140S/Q148H/G149A	13	62.0 ± 8.8
G140S/Q148H/N155H	30	63.0 ± 14.9
C56S/G140S/Q148H/G149A	10	65.1 ± 7.8
L74M/V75A/G140S/Q148H	11	53.4 ± 7.5
L74M/E138K/Q148R/R263K	7	13.1 ± 3.2
L74M/G140S/S147G/Q148	147	23.2 ± 4.5
T66I/L74M/E138K/S147G/Q148R/S230N	5	61.6 ± 15.1

**Table 5 viruses-13-00205-t005:** IN mutants selected against CAB. IN double and complex mutants that have been selected by CAB in vitro or identified in IN susceptibility assays are listed. The FCs of the potencies of CAB against the IN mutant relative to its potency against WT HIV-1 are shown, as is the single round infectivity of the IN mutant relative to WT HIV-1.

IN Mutant	FC (Potency against Mutant HIV-1 IN Relative to WT Potency)	Single Round Infectivity (Relative to WT HIV-1)
L74M/Q148R	18	42.3 ± 6.9
E138A/Q148R	11	75.2 ± 16.1
E138K/Q148K	322	49.4 ± 7.5
E138K/Q148R	10	59.9 ± 5.4
G140A/Q148K	164	12.3 ± 1.3
G140C/Q148R	28	28.8 ± 2.0
G140S/Q148K	36	44.6 ± 8.5
G140S/Q148R	173	32.6 ± 7.0
Q148H/N155H	5	7.3 ± 2.9
Q148R/N155H	21	13.8 ± 4.1
C56S/G140S/Q148H	15	69.3 ± 11.0
V72I/E138K/Q148K	136	47.4 ± 4.3
L74M/G140C/Q148R	92	29.4 ± 7.3
L74M/G140A/Q148R	22	32.5 ± 4.4
V75A/G140S/Q148H	28	56.2 ± 6.1
T97A/Y143R/N155H	59	5.2 ± 1.3
T97A/G140S/Q148H	18	55.9 ± 13.1
T122N/G140S/Q148H	49	65.0 ± 5.3
E138A/G140S/Q148H	29	85.2 ± 10.2
E138K/G140A/Q148K	255	42.9 ± 7.5
E138K/G140C/Q148R	56	44.4 ± 7.1
E138K/G140S/Q148H	39	54.4 ± 15.9
G140S/Y143R/Q148H	47	52.3 ± 1.1
G140S/Y143R/N155H	8	2.9 ± 0.8
G140S/Q148H/G149A	82	62.0 ± 8.8
G140S/Q148H/N155H	1110	63.0 ± 14.9
G140S/Q148H/G163K	14	49.6 ± 17.2
C56S/G140S/Q148H/G149A	53	65.1 ± 7.8
L74M/V75A/G140S/Q148H	109	53.4 ± 7.5
L74I/E138K/G140S/Q148R	5	17.2 ± 4.0
L74M/E138K/Q148R/R263K	44	13.1 ± 3.2
L74M/G140S/S147G/Q148K	485	23.2 ± 4.5
T66I/L74M/E138K/S147G/Q148R/S230N	76	61.6 ± 15.1

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
