# Peer review of "Integrase Strand Transfer Inhibitors Are Effective Anti-HIV Drugs"

_viruses, 2021, doi:10.3390/v13020205_

Round 1
Reviewer 1 Report
In this manuscript, the authors presented a review of integrase strand transfer inhibitors (INSTIs). Briefly, the authors described first and second generation of INSTIs, an overview about discovery, approval, efficacy, mechanism of action, their mechanism of resistance, mutation and replication and possible application as prophylaxis therapy among other applications. Overall, this manuscript has been well-written, and I think it is highly appropriate for publication in Viruses, with additional modifications. See below for what authors should complete in this manuscript.
GENERAL
Authors provided a well-organized information. Since this is an INSTIs review manuscript, current regiments of ART administration, including dose treatment, should be included as a table or as part of supplementary section. Recent reports for adverse effects and discontinuation of INSTIs perhaps can be extended in the manuscript.
Moreover, the addition of a table with EC50 values (range) use for in vitro cell culture studies would be very useful, especially in the virology field.
SPECIFIC POINTS
Figure 1. Name of each INSTIs should be listed in this figure: Raltegravir (RAL), Elvitegravir (EVG), Dolutegravir (DTG), Bictegravir (BIC), Cabotegravir (CAB).
Figures 2, 3, 4, 5 and 6. Small fond size was used in figures, which makes it difficult to read.
Tables 1, 2, 3, 4 and 5. Standardization in format an
Author Response
We have a provided a table (Supplementary table 1, referred to as Table S1 in the text) which lists the current recommended ART combination antiviral regimens that include INSTIs. The new table provides the dosing recommendation and lists the reported adverse events for the different therapies. We have also fixed all of the specific points that the reviewer listed. However, we did not provide a table with a range of EC50 values for the various in vitro cell culture studies that measured the potencies of the compounds against the INSTI-resistant mutants. There are several reasons that this kind of comparison is difficult. The most important problems come from the inherent differences in the assays used to determine the EC50s. For example, multi-round assays should show a lower EC50 than a single-round assay. It is not obvious how to compare the available data in meaningful way. Nor is it clear how showing data that are expected to be different, based on the differences in the assays, would significantly improve what has already been provided.
Reviewer 2 Report
In this manuscript,
Steven J. Smith and his co-workers discuss the current state of the clinically relevant Integrase strand transfer inhibitors (INSTIs), recommended for the first line treatment of human immunodeficiency virus type one (HIV-1) infection, and the future outlook for this class of antiretrovirals. This review provides an exhaustive and sufficiently updated analysis of the topic under investigation. It is well-written, clear and consistent, therefore once it has been subjected to minor revisions, it will be suitable for publication in Viruses.
1) Check all the Tables, unify their format, improve their resolution and introduce a title in the captions.
2) "..." should be replaced "..."
- line 436: "in the active of the HIV-1 intasome" with "in the active site of the HIV-1 intasome"
- line 442: "in the active of the HIV-1 intasome" with "in the active site of the HIV-1 intasome"
- line 530: "viral DNA stand" with "viral DNA strand"
- line 543-544: "the infectivity of the primary RAL mutants are" with "the infectivity of the primary RAL mutants is"
- line 668: "the linker the that connects" with "the linker that connects"
- line 703: "are known to reduce the in susceptibility" with "are known to reduce the susceptibility"
Author Response
We have improved and unified the tables and have added captions. All of the suggested revisions were made and we our grateful for reviewer 2 for pointing them out.